# Hypothetical Pathogenetic Model of Membranous Nephropathy

**DOI:** 10.3390/ijms26052206

**Published:** 2025-02-28

**Authors:** Irina Zdravkova, Eduard Tilkiyan, Desislava Bozhkova, Teodor Kuskunov, Yovko Ronchev, Boris Kirilov

**Affiliations:** 1Department of Propaedeutics of Internal Diseases, Medical Faculty, Medical University of Plovdiv, 4000 Plovdiv, Bulgaria; kuskuno@abv.bg; 2Nephrology Clinic, University Hospital “Kaspela”, 4000 Plovdiv, Bulgaria; eet64@yahoo.com; 3Second Department of Internal Diseases, Section “Nephrology”, Medical Faculty, Medical University of Plovdiv, 4000 Plovdiv, Bulgaria; 4Department of General and Clinical Pathology, Faculty of Medicine, Medical University of Plovdiv, 4000 Plovdiv, Bulgaria; desislava_lapteva@abv.bg; 5Department of General and Clinical Pathology, Kaspela University Hospital, 4000 Plovdiv, Bulgaria; 6Hemodialysis Unit, University Hospital “Kaspela”, 4000 Plovdiv, Bulgaria; 7Clinical Laboratory, University Hospital “Kaspela”, 4000 Plovdiv, Bulgaria; yovkoronchev@abv.bg; 8Internal Diseases Unit, Department of Emergency Medicine, University Hospital “Sveti Georgi”, 4000 Plovdiv, Bulgaria; bork07@abv.bg

**Keywords:** pathogenesis, epitope spreading, PLA2R, THSD7A, IgG abnormal glycosylation

## Abstract

Membranous nephropathy (MN) is a disease with an etiology and pathogenesis that are still not fully understood, and it represents a great challenge. It is characterized by a variable course, spontaneous remissions and relapses. The inability to rely entirely on antibodies and the continuous threat of a malignant disease make the differentiation of MN types extremely difficult. Data of twelve patients with membranous nephropathy, ranging in age between 28 and 67 years, are presented; in total, seven men and five women were observed for a period of 2 to 10 years. In all patients, the diagnosis was confirmed through kidney biopsy and laboratory tests, including immunological, histopathological, and immunohistochemical tests. Histopathological and immunohistochemical tests were applied on available material from the thyroid gland in two patients and the gallbladder in two patients with MN. Data of 102 patients with MN and their comorbidities are evaluated in order to establish correlations. These and other data are used to build a hypothetical pathogenetic model that explains the etiology and the likely pattern of disease occurrence. We found a connection between chronic cholecystitis, thyroiditis, hepatitis, and other diseases in the occurrence of MN and disease course. From our practice and cases, we drew the conclusion that chronic inflammation in sites that express PLA2R leads to the formation of antibodies against PLA2R. These antibodies occur as a preformed immune complex or separately and are deposited in the subepithelial space, leading to MN appearance.

## 1. Introduction

Membranous nephropathy (MN) is a disease with an etiology and pathogenesis that are still not fully understood. MN is one of the most common causes of nephrotic syndrome in adult patients (approximately 25%) [1,2]. It is a glomerulonephritis with immunocomplex pathogenesis, in which deposits of IgG and complement fractions are deposited on the subepithelial surface of the glomerular basement membrane and, over time, lead to its diffuse thickening. Primary MN (pMN) is a glomerular-specific autoimmune disease that accounts for about 75–80% of MN cases. It has been established that most patients with pMN have circulating antibodies for phospholipase A2 receptor (PLA2R), and the remaining cases of pMN can be classified as idiopathic MN (iMN) [3]. Secondary MN (sMN) accounts for 20–25% of cases of MN, and it is caused by autoimmune diseases (systemic lupus erythematosus, autoimmune thyroid disease, rheumatoid arthritis), infections (Hepatitis B and C, HIV, syphilis, schistosomiasis), malignant diseases, drugs and toxins. MN is characterized by a variable course and spontaneous remissions and relapses. In transplanted patients, 40% of patients develop MN of the kidney allograft and MN “de novo” [3]. In 2016, an increase in the incidence of MN was reported in China, with air pollution being assumed to be the main cause. In the first place is IgA nephropathy (28.1%), followed by MN with a frequency of 23.4% [4]. In 2017, another study conducted in China was published, where of 5935 biopsies, 4855 were diagnosed with primary glomerulonephritis. In the first place was MN (43.3%), followed by IgA nephropathy (34.1%) [5].

In 1959, Walter Heymann et al. described Heymann’s nephritis [6]. In 1978, Makker proved that the pathogenetic antigen was a mannose-containing glycoprotein, and it was subsequently called megalin [7]. Megalin is present on the surface of the epithelium of proximal tubule brush borders and that of glomerular podocytes [8].

The exact epitopes in megalin responsible for active Heymann’s nephritis are limited to only one small part of the molecule [9]. Currently, many such epitopes have been identified, and native-like conformation and glycosylation are essential for the induction of the disease [10,11].

In 2009 Beck et al. described the role of anti-PLA2R antibody (APLA2R), in the pathogenesis of a very large proportion (≥70–80%) of iMN patients [12].

In 2014, Nicola M. Tomas and Laurence H. Beck et al. established a new antigen present, mostly in patients with iMN who are negative for APLA2R. The isolated antigen is thrombospondin type-1 domain-containing 7A (THSD7A), localized on podocytes [13].

The frequency of anti-PLA2R1-associated MN appears to be lower, about 53%, in Japan compared with other countries [14,15]. The reported incidence of THSD7A-associated MN ranges from 5.5% to 6.1% [13,16]. Positive antibodies against PLA2R1 and against THSD7A are detected in 1% of patients with MN [16].

Sanjeev Sethi et al. contributed to the discovery of new antigens: exostosins 1 and 2 (EXT1/2), neural epidermal growth factor-like 1 protein (NELL-1), semaphorin 3B (Sema3B), and neural cell adhesion molecule 1 (NCAM1) [17]. However, some of these new antigens are characteristic of patients with systemic lupus, a pediatric population, and a large percentage are associated with the presence of malignant disease [18].

In 2007, Pallavi Shah et al. described immunization with megalin which has four distinct ligand-binding domains (LBDs) that contain epitopes to which antibodies are directed. Serum from rats 4 weeks after immunization with L6 demonstrated activity for only the L6 fragment, whereas serum from the same rats 8 weeks later demonstrated reactivity with all four recombinant fragments (L6 and LBDs II, III, and IV). They demonstrated that the L6 immunogen does not contain the epitopes responsible for reactivity to the LBD fragments. Therefore, the appearance of antibodies directed against LBD fragments several weeks after the primary immune response suggests epitope exposure [19].

Seitz-Polski et al. identified distinct epitopes in PLA2R within CysR, CTLD1, and CTLD7 that provoke reactivity against anti-PLA2R antibodies, affirming “intramolecular epitope spreading” in MN in humans [20]. CysR is the primary epitope, and it spreads to CTLD1 and CTLD7. Anti-PLA2R1 reactivity against CysR is linked to favorable outcomes, while reactivity against CTLD1 and CTLD7 correlates with disease activity. Reinhard et al. identified a novel epitope in the CTLD8 domain [21].

Hong Tang et al. observed that patients with newly diagnosed pMN carry two populations of PLA2R-Abs in sera. PLA2R-Ab1 appeared at an earlier time point, whereas increased levels of PLA2R-Ab2 coincided with the worsening of proteinuria [22].

In sMN, positive antibodies for APLA2R have been found in sarcoidosis, lupus nephritis, and hepatitis B [23]. Several cases of THSD7A expression by tumor cells have also been described [1]. In a recent systematic review on APLA2R positivity, detected by indirect immunofluorescence, APLA2R was detectable in 73% of patients with pMN and 14% with sMN [24]. Studies have analyzed cases of sMN in patients with hepatitis B, hepatitis C, sarcoidosis, and malignancies and found positive staining for PLA2R in 64% (25 of 39), 64% (7 of 11), 75% (3 of 4), and 70% (7 of 10) of cases, respectively [16,24,25].

In December 2020, George Haddad et al. published a study demonstrating that impaired glycosylation of IgG4 causes the activation of the lectin complement pathway in patients with APLA2R1-associated MN and abnormalities in glycosylation are correlated with the severity of the disease [26]. They relate this to data from Malhotra et al., who found that IgG autoantibodies lacking the terminal galactose residue at position Asn 297 of the constant fragment (Fc) were able to bind and activate MBL [27].

Chronic inflammation is characterized by numerous systemic physiological and biochemical changes, most of which are mediated by abundantly secreted proinflammatory cytokines and is characterized by marked changes in glycosylation. Glycosylation plays an important role in a variety of biological functions, including protein stability and effector functions, intercellular interactions, signal transduction, and cell immunogenicity [28].

The impossibility of completely relying on antibodies and the continuous threat of a malignant disease make the work of clinical nephrologists extremely difficult and a matter of great responsibility.

## 2. Results

### 2.1. Data from Patients with MN and Their Comorbidities

We present data from 102 patients, 57 men and 45 women with MN, whose medical records were reviewed retrospectively to establish a relationship between comorbidities and MN. The patients are divided into three groups: pMN, iMN, and sMN, based on PLA2R positivity as well as clinical and paraclinical evaluation. Arterial hypertension was not included in the study. If the number of one comorbidity was not exceeding five patients (for this study), it was considered as non-relevant from a hypothetical pathogenetic point of view. We decided to group the comorbidities into four main groups according to the organ from which they originate. One of the frequent comorbidities were the diseases of the gall bladder; here were included cholecystitis (with or without polyps in the gall bladder) and patients with cholecystectomy and cholelithiasis. Another frequent group of comorbidities were the thyroid diseases: autoimmune thyroiditis, nodose toxic goiter, patients with partial thyroidectomy and total thyroidectomy. In the third group were lung diseases: bronchial asthma, pulmofibrosis, bronchiectasis, tuberculosis, COPD for more than five years, two patients with acute viral infection preceding the MN (one with haemoptysis and one with pneumothorax), and lung cancer. In the last group were the liver diseases: toxic hepatitis, hepatitis B (no patients with hepatitis C), and one patient with cholangiocarcinoma. Another common comorbidity was diabetes mellitus, but data on it are not included here. If a specific comorbidity was observed in five or fewer patients and could not be included in the main groups, it was considered irrelevant from a hypothetical pathogenetic point of view. The comorbidities not included as relevant in this study and the number of patients: Psoriasis: 1; NSAID: 5; Kidney cancer: 1; Behcet disease: 1; Glaucoma: 1; Pancreatitis: 1; Thrombocytosis: 1; Phlegmon: 1; SLE: 1; Rheumatoid arthritis: 2; Urinary bladder cancer: 2; Myelofibrosis: 1; Hodgkin lymphoma: 1; Addison disease: 1; Thromboembolic event: 1; Steatohepatitis: 2; Prostate cancer: 1. Drug allergies were not included.

#### 2.1.1. Patients with pMN

We identified 69 patients with pMN, of whom 14 patients had comorbidities associated with the thyroid gland, 10 with the gall bladder, 9 with the liver, 15 with the lungs. Additionally, 21 patients’ comorbidities were considered non-relevant to the study or they had no comorbidities at all. As a result, 69% of patients have/had a concomitant inflammatory disease in a site expressing PLA2R. Patients’ results are presented in Table 1.

#### 2.1.2. Patients with iMN

We identified 23 patients with iMN, of whom 4 patients had comorbidities associated with the thyroid gland, 7 with the gall bladder, 1 with the liver, 9 with the lungs. Additionally, 2 patients’ comorbidities were considered non-relevant to the study or they had no comorbidities at all. As a result, 91% of patients have/had a concomitant inflammatory disease in a site expressing PLA2R. Patients’ results are presented in Table 2. The four patients with thyroid comorbidity included in the iMN group had undergone partial thyroidectomy due to a toxic node or had hypothyroidism diagnosed after MN. The patient with hepatitis was classified as having toxic, non-viral hepatitis.

#### 2.1.3. Patients with sMN

We identified 10 patients with iMN, of whom 1 patient had comorbidity associated with the liver and 1 with the lungs. Additionally, 8 patients’ comorbidities were considered non-relevant to the study or they had no comorbidities at all. As a result, 20% of patients have/had a concomitant inflammatory disease in a site expressing PLA2R. Patients’ results are presented in Table 3. In those 10 patients, MN was secondary due to: Lung cancer: 1; SLE: 1; Rheumatoid arthritis: 2; Urinary bladder cancer: 1; Hepatitis B: 1; Myelofibrosis: 1; Hodgkin lymphoma: 1; Prostate cancer: 1; Cholangiocarcinoma: 1.

### 2.2. Patient Data Supporting the Main Hypothesis

We present a few cases of MN with our interpretation of the pathogenesis and primary etiology of the disease, based on long-term observation and full immunological and immunohistopathological examination. Data on patients, including the time of disease onset relative to the primary cause as well as information regarding their positivity for antibodies in serum and from IHC are presented in Appendix A.

#### 2.2.1. Patient № 1

The patient underwent cholecystectomy one year before the appearance of MN due to stones in the gallbladder and ductus choledochus—their occurrence and duration are not known. IHC of the kidney biopsy was negative for APLA2R at the start but positive for MBL and IgG4, so it was assumed that the patient had iMN. Two years after the onset of the disease, the patient became positive for APLA2R in serum. At that time, he did not receive pathogenetic treatment. The start of diabetes mellitus coincided with the start of MN. Our understanding is that chronic inflammation in the gallbladder led to the formation of antibodies against PLA2R, since cholangiocytes also express PLA2R on their surface. IgG4, which is possibly abnormally glycosylated due to chronic inflammation and diabetes, activates the lectin pathway. Activation of the lectin pathway in the glomerulus leads to conformational changes in PLA2R, recognition from APLA2R antibodies, and epitope spreading.

#### 2.2.2. Patient № 2

The patient underwent a total thyroidectomy due to a nodular goiter two years before the appearance of MN. At the start of the disease, the APLA2R antibodies in serum were not examined (not available at that time), and after two years of treatment, the patient tested positive; at that time, the patient was on immunosuppressive treatment. When we performed IHC on the thyroid gland, it tested positive for deposition of anti-PLAR2 antibodies (Figure 1).

We consider this case of MN as pMN, which appeared due to a nodose goiter. Chronic autoimmune inflammation in the thyroid gland leads not only to the formation of anti-TPO, anti-Tg, and anti-TSHR but also to the formation of antibodies against PLA2R, since it is expressed by thyroid glandular cells. Recognition from the immune system that the kidney is the other site expressing PLA2R happens only when we have a conformational change in the structure of PLA2R on the surface of podocytes induced by the activation of the lectin pathway, which is activated due to abnormally glycosylated immunoglobulins. Epitope spreading appears on the surface of PLA2R, and a new disease entity appears—MN.

#### 2.2.3. Patient № 3

The patient presented at the beginning of MN with low viral hepatitis B replication, and pathogenetic treatment was started only after the initiation of Lamivudine. Hepatitis is characteristic of sMN, but the positive IHC results and anti-PLA2R antibodies spoke in favor of pMN, and we had very good results after the treatment. We present this case because we believe that it was not a simple coincidence of hepatitis and pMN. Chronic viral hepatitis led to the deposition of hepatitis Ag in the subepithelial space, and chronic inflammation led to the abnormal glycosylation of IgG4 directed against the primary antigen, which caused the activation of the lectin pathway, conformational changes in PLA2R, epitope spreading, and the appearance of MN. It is possible that the antibodies against PLA2R were already preformed because hepatocytes express PLA2R on their surface, and, in the course of hepatitis, chronic inflammation leads to the abnormal glycosylation of IgG4, which activates the lectin pathway and causes conformational changes in the structure of PLA2R on the surface of podocytes and epitope spreading.

#### 2.2.4. Patient № 4

A patient diagnosed with diabetes mellitus type 2 six months before the kidney biopsy. The patient had no history of known HBV infection and was HBsAg-negative before the kidney biopsy. It was estimated that the patient had iMN, and immunosuppressive treatment was started. After six months of treatment, the patient still had pronounced nephrotic syndrome and suffered one thromboembolic event and coronary artery stent placement. He underwent many new investigations; positive HbcAb-33.65 (n.range < 0.9) and low-grade HBV-DNA replication were registered, so treatment with Lamivudine was started in the background of the same pathogenetic treatment. These actions led to full clinical and paraclinical remission and a slow reduction of treatment, with the first withdrawal of corticosteroids and pulse treatment with cyclophosphamide at six months, as well as the withdrawal of azathioprine after one year. For four years, there were no relapses, and pathogenetic treatment was no longer required two years after he stopped the treatment with Lamivudine. The course of the disease indicated sMN, which was corroborated by the negative APLA2R and MBL results from IHC. We present this case due to our understanding that subepithelial hepatitis antigen deposition in the background of diabetes mellitus leads to the development of MN; in this case, complement activation happened through the classical pathway due to the subepithelial deposition of antibodies against hepatitis antigens, but the damage to the kidney was triggered by the appearance of diabetes mellitus.

#### 2.2.5. Patient № 5

The patient was admitted with high-grade proteinuria and edema, as well as the following concomitant diseases: psoriatic arthritis treated with methotrexate and a recently established HBV infection, for which the viral replication results were still expected. Viral replication was determined to be positive, and treatment with Lamivudine was started, which led to the spontaneous remission of MN. Chronic inflammatory processes led to conformational changes in PLA2R and partial epitope spreading, but not enough to form a new disease entity, since treatment with Lamivudine was started. As we see in this case, MBL was negative, which may have been due to the advanced stage of MN, since it was found to be positive in the earlier stages. The antibodies against PLA2R were possibly preformed due to hepatitis, and PLA2R was expressed by hepatocytes, but the structure of PLA2R did not suffer severe conformational changes and full epitope spreading. This is why we did not have an increase in anti-PlA2R antibodies in serum. In the material from the kidney biopsy, slight focal and segmental glomerular changes were present as well as interstitial nephritis, which we believe resulted from the patient’s treatment for psoriatic arthritis—Methotrexate and Salazopyrin. This was an additional factor facilitating kidney damage.

#### 2.2.6. Patient № 6

The sixth patient was admitted for general surgery with phlegmon on the big toe of his left foot and nephrotic syndrome. He underwent amputation of the big toe; Proteus and Pseudomonas were isolated from the wound, and diabetes type 2 was diagnosed. After amputation, he was admitted to our clinic, where we continued the antibiotic treatment and performed a kidney biopsy. The patient had no history of known HBV infection and was HBsAg-negative before kidney biopsy, HbcAb-199.29 (n.range < 0.9), HbeAg 0.02 (negative), and HbeAb- 40.0 (n.range < 10 negative). After consultation with a gastroenterologist, it was considered that he suffered from HBV infection, which had resolved on its own. The patient was negative for PLA2R antibodies and THSD7A at the start of MN. From the IHC and serum antibody results, he was classified as having iMN (Figure 1). One year after the start of pathogenetic treatment, the result for THSD7A became positive. He was treated only once at the start with 500 mg of methylprednisolone for three days in a row and 500 mg of cyclophosphamide four times over one year. After that, the pathogenetic treatment was stopped and never repeated. The patient confirmed chronic alcohol abuse, which we had suspected. His visits were irregular, and he frequently attended only due to our insistence. After three years of absence, he returned, and a lesion in the liver was discovered. It was verified via biopsy to be moderately differentiated cholangiocarcinoma. From this point of view, we can classify this MN as secondary, but the fact is that after one year of partial treatment, he was in full clinical and paraclinical remission for three years. Hepatocytes also express THSD7A on their surface. Sensitization to podocyte THSD7A happens in the background of diabetes mellitus and chronic alcoholism, and podocytes also release IL1, causing the abnormal glycosylation of IgG4, as well as the activation of complement, so THSD7A becomes exposed, and antibodies against it become actively produced.

#### 2.2.7. Patient № 7

The patient presented with non-nephrotic-range proteinuria, high blood pressure, skin photosensitivity to the sun, chronic cholecystitis, and type 2 diabetes mellitus, which were diagnosed one year before the appearance of MN. The patient was positive for PLA2R antibodies in serum and for anti-PLA2R antibodies according to the IHC results. The immunofluorescence from the kidney biopsy demonstrated the following: 3–4/+/subepithelial granular and pseudolinear deposition of IgG, IgM, and C3; 2–3/+/of IgA and C1q- with the same character and localization, and very weak C4. The disease was classified as sMN in systemic lupus erythematosus (SLE). After six months of treatment, she was negative for ANA and anti-dsDNA and positive for antibodies for antiphospholipid syndrome: APL IgG 11.95 (reference range < 10 GPL); APL IgM 62.59 (reference range < 10 MPL); ACL IgG 11.91 (reference range < 10 GLP); ACL IgM 53.70 (reference range < 7 MPL); anti-β2 GP IgM 43.68 (reference range < 8 E/mL); anti-β2 GP IgG < 8 (reference range < 8 E/mL). We present this case in order to underline the fact that positivity for PLA2R in serum is equivalent to the diagnosis of MN, but it is not sufficient for the categorization of the type of MN that the patient has, and this is an indicative case of PLA2R-positivity in sMN. Chronic cholecystitis led to the formation of antibodies against PLA2R, and the presence of diabetes mellitus with other SLE antibodies that underwent subepithelial deposition led to the activation of the lectin pathway, conformational changes in the structure of PLA2R, and epitope spreading. After two years of treatment, the patient was in full clinical and paraclinical remission and was redirected to a rheumatologist for treatment and observation.

#### 2.2.8. Patient № 8

The patient was first admitted to endocrinology with thyrotoxicosis and positive results for anti-Tg and anti-TPO, and was diagnosed with Graves’ disease and concomitant nephrotic syndrome. Thyroidectomy was performed, and after three weeks, a kidney biopsy was performed. IHC was not performed on the kidney biopsy because the patient had a kidney biopsy in another clinic. PLA2R in the serum was positive, and pathogenetic treatment was started. IHC was performed on the material from the thyroidectomy. (Figure 1).

In our understanding, in autoimmune thyroiditis, the destruction of the thyroid gland leads to the exposure of PLA2R in thyrocytes and the formation of antibodies against it. On the other hand, immune complexes among TPO, Tg, TSHR, and antibodies against them can be deposited in the subepithelial space. This can lead to release of IL1 from the podocytes, causing the abnormal glycosylation of Ab, the activation of the lectin pathway, conformational changes in PLA2R, and epitope spreading. This process is slow, and this is why thyroiditis is always described and associated with kidney diseases. It is possible that abnormal glycosylation does not happen in the kidney. Usually, the antibodies after total thyroidectomy disappear over time, but not in our case. The overall time of surveillance for our patient was five years; the patient had a total thyroidectomy (even the parathyroid glands suffered during the operation) and remained positive for anti-TPO and anti-Tg, which even correlated with PLA2R serum antibodies and disease activity. It is very possible that they have a structural resemblance.

#### 2.2.9. Patient № 9

The patient presented with nephrotic syndrome and was slightly positive for PLA2R in serum, and the result from kidney biopsy confirmed MN. Pathogenetic treatment started with good results. Interestingly, the patient had a cholecystectomy 6 years before the appearance of MN. Material from the gallbladder was used for IHC (Figure 1).

We consider that chronic cholecystitis led to the formation of antibodies against PLA2R (PLA2R expressed by cholangiocytes). These abnormally glycosylated antibodies (possibly at the primary site, the gallbladder) activated the lectin pathway in the subepithelial space, causing conformational changes in PLA2R on the surface of podocytes and leading to the formation of a new disease entity: MN.

#### 2.2.10. Patient № 10

The patient presented with mild nephrotic syndrome and easily entered remission within one month after an infusion of Cyclophosphamide 500 mg. They were treated with Azathioprine 50 mg daily for one year, and two years after, they remained in full remission. The patient had chronic cholecystitis and cholecystectomy 9 years before the appearance of MN and Hashimoto thyroiditis, which was diagnosed at the same time as MN. PLA2R in serum was negative. The material from the cholecystectomy was available, and IHC performed (Figure 1).

In this case, we consider that chronic inflammation in the gallbladder led to the formation of antibodies against PLA2R, but MN appeared together with Hashimoto thyroiditis. This led us to the conclusion that, in some cases, chronic inflammation in only one primary site is not enough for the appearance of MN. The trigger can be diabetes mellitus, Hashimoto thyroiditis, or severe inflammation in another primary site in which the cells also express PLA2R.

#### 2.2.11. Patient № 11

The patient was positive for APLA2R in serum at the start of MN and had autoimmune thyroiditis for more than 10 years before MN. This patient had a very good response to pathogenetic treatment and entered stable remission easily. Interestingly, the level of anti-TPO in serum at the start of the disease was 342, and that of antibodies against PLA2R was 57. One year after treatment, her anti-TPO level decreased to 97, and the level of antibodies against PLA2R in serum increased to 188. At that time, the patient was in full clinical and paraclinical remission (proteinuria 0.49/24 h, normal albumin, and total protein in serum). We believe that the test for PLA2R in serum registered antibodies against anti-TPO instead, maybe they have a structural resemblance. After a time, both of them decreased to normal values.

#### 2.2.12. Patient № 12

The patient first presented ten years ago with severe nephrotic syndrome. Six and eight months before the appearance of MN, he was admitted to the surgical ward twice due to hemoptysis. A bronchoscopy was performed and revealed hyperemic, edematous, easily bleeding mucosa, mostly in the right upper bronchi, but no direct source of bleeding was found. A lung scan was performed, which revealed data on old pleural adhesions on the left and pleuropneumonic changes with pleural effusion on the right, and the findings did not have characteristics of a neoplastic process. In the control scan, the findings on the right had undergone complete fraternal development, and the pleural adhesions persisted, with no data on enlarged lymph nodes and pleural effusions. The scans were performed before the second hemoptysis. The patient worked as a construction builder at the time that MN appeared. The patient had pMN, which we believe was due to air pollution. We considered PM2.5 as the main etiological factor, and the hemoptysis was caused by a superimposed pulmonary infection. In that case, we observed a switch from one antibody to another; according to IHC, the patient was positive for antibodies against PLA2R and was positive for THSD7A in serum later in the course of the disease when we tried to stop the pathogenetic treatment. Interestingly, positivity for THSD7A occurred one year after the appearance of diabetes mellitus. PLA2R is expressed by podocytes and NK cells in the lungs, and THSD7A is expressed by podocytes and type 1 alveolar cells. We are not completely sure why we had patients who were negative for PLA2R in serum but positive according to IHC (of 45 patients with pMN, 4 were negative for PLA2R antibodies in serum). They may have had atypical epitope spreading, and the antibodies could not be detected with standard methods. We believe that the disease started in the lungs and led to the sensitization of antibodies against PLA2R; since the patient continued working in construction, this led to sensitization against THSD7A. Interestingly, positivity for THSD7A was not associated with the recurrence of nephrotic syndrome but only with an elevation in proteinuria, which, after one month of treatment with Azathioprine, went back to normal without corticosteroid treatment. For the last four and half years the patient has been in full remission without further treatment. Our explanation is that the patient’s change in occupation may have contributed to this remission.

### 2.3. Hypothetical Pathogenetic Model of Membranous Nephropathy

The chronic inflammation at a site of extrarenal expression of PLA2R causes changes within its structural conformation and the following sensitization of the immune system against it, resulting in the production of anti-PLA2R IgG4. Normal structure of PLA2R is presented on Figure 2.

Antigens from the primary site and the aforementioned antibodies circulate around the body and eventually find their way into the glomerulus, where they are deposited subepithelial. This could happen under the form of preformed immune complexes or newly created immune complexes, following the separate deposition of an antigen and antibodies. Interaction between PLA2R and antibodies is presented on Figure 3.

Normally, IgG4 within these immune complexes is unable to activate the complement system. However, under certain circumstances, it is liable to modifications. IgG4’s capacity to bind MBL can be increased through abnormal glycosylation, particularly through the loss of galactose in the glycans of the antibody’s Fc-region.

This could happen in several ways:(1)before the immunoglobulins have been deposited in the kidney—at the site of primary chronic inflammation;(2)after having been deposited in the subepithelial space—under the influence of IL-1, secreted by the podocytes.

Both pathways can be influenced by the presence of diabetes mellitus.

The abnormally glycosylated (lacking galactose) IgG4 antibodies are now capable of activating MBL. MBL, in turn, initiates the alternative complement pathway, leading to conformational changes in the structure of PLA2R expressed by the podocytes. The newly altered structure of podocyte PLA2R now matches that of the one in the primary site of chronic inflammation. The already formed anti-PLA2R can bind to it and initiate renal pathology. The prolonged inflammatory process leads to epitope spreading and the additional recognition of other segments of the renal PLA2R molecule as potential targets for the immune system, further amplifying the inflammatory response in the kidney. Hypothetical pathogenetic model is presented on Figure 4.

Immune complexes that are preformed or formed in situ can contain foreign antigens (hepatitis antigens, thyroid antigens, tumor antigens) or can contain PLA2R from another primary site (cholangitis, thyroiditis, and cancers of the skin, breast, gastrointestinal tract, lung, etc.).

## 3. Discussion

PLA2R1 is a receptor for secretory phospholipase A2 (sPLA2), and the PLA2R gene is located on Chromosome 2, Cytoband q24.2. Polymorphisms at this locus have been associated with susceptibility to idiopathic membranous nephropathy [29,30].

The cell type enrichment of the selected gene in all non-core cell types is found in the following, in descending order: podocytes; eccrine sweat gland cells; peritubular cells; breast myoepithelial cells; breast glandular cells (progenitors); minor salivary glandular cells; breast glandular cells; ductal cells; mesothelial cells; Leydig cells; intercalated cells; adipocytes (skin); mitotic cells (heart); thyroid glandular cells; Sertoli cells; cardiomyocytes; undifferentiated cells (pituitary gland); adrenal cortex cells; cholangiocytes; alpha cells; enteric glia cells; outer root sheath cells; thyrotropes; minor salivary gland ductal cells; NK cells (lungs) [29,30].

From our patient data, the most frequent comorbidities in pMN and iMN are located in sites which express PLA2R. Even malignancies linked with sMN originate from sites expressing PLA2R. For example, we had a patient with Addison’s disease who was positive for PLA2R, and another with skin cancer who also tested positive for PLA2R. Diseases cannot be strictly categorized or confined to predefined frames;, they just exist, adapt, and evolve, and so does our response to them.

From our practice and cases, we drew the conclusion that chronic inflammation at sites that express PLA2R can lead to the formation of antibodies against PLA2R; these antibodies occur as a preformed immune complex or separately and are deposited in the subepithelial space. However, this alone is insufficient; we need conformational changes in the structure of PLA2R and further epitope spreading in order to establish MN as a new distinct disease entity. Epitope spreading is a phenomenon that occurs after the primary immune response. When the immunodominant response fails to clear the target, the immune system mounts a broad immune response against different epitopes on the same molecule or on different molecules [19]. If epitope spreading is complete, the disease evolves; if epitope spreading is partial, interventions like cholecystectomy or thyroidectomy can stop the process and lead to remission. This is why Hong Tang et al. observed two populations of PLA2R antibodies: one that appears early and another that coincides with the worsening of proteinuria [22]. This explains spontaneous remission and the fact that, in some patients, positivity against anti-PLA2R antibody from the primary site can be detected long before the onset of MN.

PLA2R from the primary site after thyroiditis, cholecystitis, hepatitis, or chronic lung disease undergoes conformational changes and epitope spreading, but PLA2R in the kidney is still normal. The immune system recognized the kidney as another site expressing PLA2R only if conformational changes occur in the structure of PLA2R on the surface of podocytes. These changes are induced by the activation of the lectin pathway, which is activated by abnormally glycosylated immunoglobulins or additional triggers.

One such trigger is diabetes mellitus. In diabetes, we have thickening of the GBM and glycation of structural proteins which reduce the anionic filling, allowing the transition and deposition of immunoglobulins in/on the surface of the GBM. Other triggers include aggravation of inflammation in the primary site and the onset of inflammation in another primary site that also expresses PLA2R (chronic cholecystitis and superimposed autoimmune thyroiditis). The intake of NSAID is another potential trigger. There is growing evidence supporting the role of NSAIDs in the induction of MN. Nasrallah et al.’s studies in diabetic rats evaluated the kidney effects of NSAIDs. They found that NSAIDs reduce GBM (glomerular basement membrane) thickness and slit pore diameters, decreasing the density of podocytes and increasing the mesangium [31]. We believe that these changes make PLAR disposable.

We cannot comment on the exact means of interaction on the molecular level, but it is clear that the lectin pathway plays a large role in the pathogenesis of MN, especially in the early stages, where it is most probably activated by abnormally glycosylated immunoglobulins. These immunoglobulins are altered due to chronic and continuous inflammation at the primary site or in situ, with the help of podocytes, which, in their effort to overcome inflammation, secrete IL1. In that way, podocytes also contribute to the onset of the disease.

We believe that chronic subepithelial inflammation is caused by foreign antigens (hepatitis antigens, thyroid antigens, tumor antigens, and immune complexes in SLE) that are covered by podocytes with protruding PLA2R and other podocyte antigens. This process leads to the activation of complement, conformational changes in the structure of podocyte antigens, the formation/sensitization of antibodies against them, and the appearance of MN. That pathogenetic model can be applicable not only to PLA2R but also to other antigens on the surface of podocyte. This explains the broad spectrum of antibodies discovered in MN—EXT1/2, NELL-1, Sema3B, and NCAM1—as well as cases of double positivity for PLA2R and THSD7A, and their appearance in pMN and sMN [16,17,18,24,25].

Our aim was to present a hypothetical model which explains this diversity of antibodies observed in MN. This model integrates much of what is discovered until that moment about MN wile also incorporating connections to chronic lung diseases and air pollution, as described by Wenbin Liu et al. in 2019. In their hypothetical model, they proposed a potential link between inflammation, pollution, and PLA2R [32]. This is also consistent with the increasing frequency of MN, for which air pollution has been suggested as the main cause of increased incidence of MN [4]. The existing MN classifications that we have are functional and applicable, and the hypothetical model builds upon them while raising questions that warrant further investigation: Is epitope spreading complete? What factors facilitate it? Can the eradication of inflammatory process at the primary site possibly halt disease progression?

## 4. Materials and Methods

### 4.1. Materials

Group I—data from 102 patients with membranous nephropathy, aged between 27 and 86 years (57 men and 45 women), treated at the Nephrology Clinic of the University Hospital “Kaspela” for the period December 2012 until March 2023, are used for evaluation of comorbidities. Inclusion criteria: patients with MN whose diagnosis is proved by kidney biopsy. Exclusion criteria: lack of MN from kidney biopsy.

Group II—data from 12 patients with membranous nephropathy, aged between 28 and 67 years (in total, 7 men and 5 women), treated at the Nephrology Clinic of the University Hospital “Kaspela” for a period of 2 to 10 years. In these patients, the onset of MN was examined in detail, a temporal relationship was established between the onset of MN and the underlying disease responsible for the appearance of MN. In patients with available material from the primary site (in our cases, the gallbladder and thyroid gland), IHC was performed.

In both groups, the diagnosis was confirmed using puncture kidney biopsy and laboratory tests, including immunological, histopathological, and immunohistochemical tests. Before undergoing kidney biopsy, patients were tested for HBV, HCV, and HIV. If we had medical indications, tumor markers were also examined. All patients had a biopsy-proven diagnosis of MN. Testing for anti-PLA2R antibodies was performed, although in some cases, it was conducted at the later stages of the disease, since these tests were not available at the beginning. In such cases, IHC for anti-PLA2R antibodies was performed on kidney biopsy material. Thrombospondin antibodies were available for a limited period of time, so they were not studied in all patients.

### 4.2. Methods

#### 4.2.1. Immunohistochemical Testing

Immunohistochemical testing was performed in accordance with the manufacturer’s standard protocols. The following antibodies were used: recombinant anti-PLA2R anti-bodies [EPR20483] (ab211573), anti-IgG4 antibodies (ab232869), and anti-Mannan-binding lectin/MBL antibodies [3B6] (ab23457). All were from the company “Abcam PLC[M44][IZ45]” (152 Grove Street, Waltham, MA, USA).

Serial sections that were 4 μm thick were prepared from paraffin blocks and mounted on adhesive slides. The sections were deparaffinized and rehydrated in alcohols of decreasing concentration. Washing was carried out with Bond TM Wash Solution [M46][IZ47] from the company Medical Technology Engineering, Ltd. (Sofia, Bulgaria) according to the instructions for use. Prior to performing the immunohistochemical reaction, heat-mediated antigen retrieval was performed by incubating the sample in Bond TM Epitope Retrieval Solutions 1 and 2 with a pH 9.0 buffer.

Serial sections from each of the studied cases were tested for the antibodies used. Positive and negative controls were prepared for each run of the antibody assays. The positive control was chosen according to the manufacturer’s instructions; namely, for anti-PLA2R, peritumoral glomeruli around clear cell carcinomas were used; for anti-IgG4, prostate carcinomas were used; for anti-MBL, hepatocellular carcinomas were used.

Immunohistochemistry was performed according to the manufacturer’s instructions using the Bond Polymer Refine Detection Kit imaging system from the company Medical Technology Engineering, Ltd. (Sofia, Bulgaria).

The negative control for each antibody was prepared using a standard immunohistochemical procedure without instilling the test antibody. The antibodies used during IHC staining were diluted; the dilution for the anti-PLA2R antibody was 1/2000; for the anti-Mannan-binding lectin/MBL antibody, the dilution factor was 1/250; for the anti-IgG4 antibody, the dilution factor was 1/1000.

An Olympus light microscope (No. OD82685) provided by the Plovdiv Medical University, 4000 Plovdiv, Bulgaria was used. In each case, at least 5 fields were selected in the serial sections, and they were observed at 400× magnification (eyepiece: ×10; objective: ×40).

Interpretation of anti-PLA2R antibody staining results: Two false-positive patterns could be observed. The first was characterized by the presence of a weak linear expression localized to the outer surface of the glomerular loop, which was observed in both the normal kidney and the negative internal controls. In the second pattern, anti-PLA2R showed a more intense “smudge” staining in the Baumann space, which was possibly due to the presence of normally expressed PLA2R protein on the podocyte membrane. In contrast to the above expression, a dot-like subepithelial granular pattern was accepted as the only true positive result. The degree of staining was scored as follows: (1+) weak expression, (2+) moderate expression, and (3+) strong expression (Figure 5).

The lack of IgG4 IHC expression in the normal kidney represented a complete lack of staining in the negative cases and a similar “dot-like” pattern in the positive cases, making interpretation easier.

Using a similar method, anti-MBL expression was explored. A complete absence of staining in the negative cases was found, and a “dot-like” pattern was observed in the positive cases.

In cases with advanced disease or extensive segmental sclerosis, IHC positivity was limited to the area of the glomeruli that had not undergone fibrosis, representing a possible cause of false-negative results.

#### 4.2.2. Immunological Testing

##### Immuno-Enzymatic Method for Quantitative Measurement of Anti-PLA2R1 Antibodies in Biological Samples (Serum and Plasma)

For the quantitative measurement of anti-PLA2R1 antibodies, a test set from the company EUROIMMUN, Germany (cat. № EA 1254-9601 G), was used for the immuno-enzymatic determination of anti-PLA2R1 antibodies in serum or plasma. The principle was immuno-enzymatic analysis based on the ELISA technique. Preliminary preparation of samples for analysis: Patient samples were prediluted at a ratio of 1:101 with sample dilution buffer. Calibrators and controls are not diluted. For qualitative or semi-quantitative analysis, only calibrator 2, positive and negative controls, and the patient sample were incubated. Calibrators 1 to 5, positive and negative controls, and the patient sample were incubated for quantitative analysis. An undiluted patient sample that was not currently being tested could be stored at 2 °C to 8 °C for 14 days. Reference limits: Sera from 191 healthy subjects were tested with EUROIMMUN ELISA. The mean concentration of antibodies binding PLA2R1 was 0.4 RU/mL, with a range of 0.0 to 5.0 RU/mL. EUROIMMUN defines a cutoff value for anti-PLA2R1 antibodies of 20 (RU/mL). Of the healthy individuals, 0% were positive. When interpreting the results, the following values are recommended:

<14 RU/mL: negative;>14 to <20 RU/mL: borderline;>20 RU/mL: positive.

When the measured values from the two determinations differed from each other, the analysis was repeated.

##### Immunofluorescence Method for Qualitative and Semi-Quantitative Determination of Anti-THSD7A Antibodies in Biological Samples (Serum and Plasma)

For the qualitative determination of anti-THSD7A antibodies, a test set from the company EUROIMMUN, Germany (cat. № FA 1254-1003-51) was used for the indirect immunofluorescence determination of anti-THSD7A antibodies in serum or plasma. The principle was an indirect immunofluorescence assay. The concentrations of anti-THSD7A antibodies were reported on an epifluorescent microscope from OMAX (model: M837ZFLP-TP) provided by Kaspela University Hospital. Preliminary preparation of samples for analysis: It was recommended to dilute the samples for semi-quantitative analysis, and PBS-Tween 20 was used for this purpose. Example of 1:10 dilution: 11.1 μL of serum was mixed with 100 μL of PBS-Tween 20 solution and mixed gently.

Reference limits: Titer 1: <10 (IgG).

The reagent manufacturer studied the prevalence (presence) of anti-THSD7A in 218 healthy individuals and found it to be 0%.

## 5. Conclusions

Our understanding of membranous nephropathy has undergone serious changes. This disease is secondary in nature if we discuss its way of appearing and pathogenesis. It appears as a consequence of another chronic illness, which can be autoimmune (SLE, thyroiditis), inflammatory (chronic cholangitis, chronic hepatitis, syphilis, schistosomiasis), paraneoplastic when the primary site expresses PLA2R (breast, lung, gastrointestinal tract, and skin cancers, etc.), or other (sarcoidosis). The treatment and eradication of primary site inflammation lead to spontaneous remission, a quicker response to pathogenetic treatment, and total eradication of the disease. However, if epitope spreading occurs and the etiological site has not been identified, it should be managed as pMN because, at one point, the disease separates as a new disease entity.

## Figures and Tables

**Figure 1 ijms-26-02206-f001:**
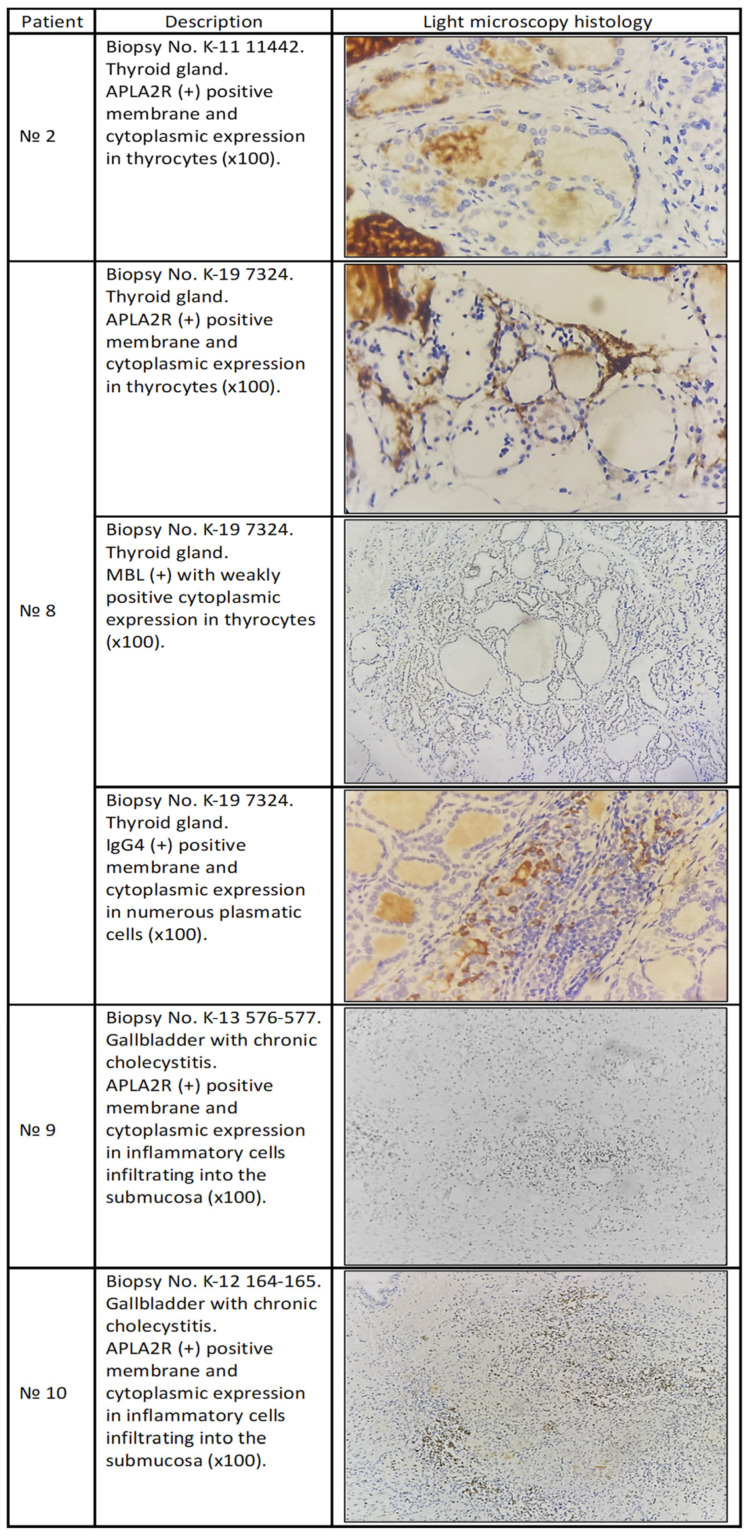
Light microscopy histology from patient № 2, № 8, № 9, and patient № 10.

**Figure 2 ijms-26-02206-f002:**
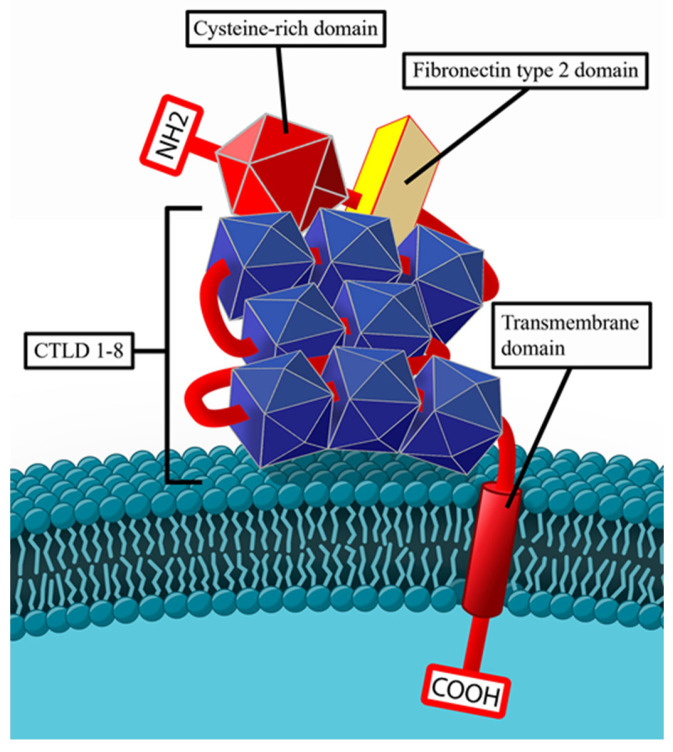
Normal structure of PLA2R on the surface of a podocyte.

**Figure 3 ijms-26-02206-f003:**
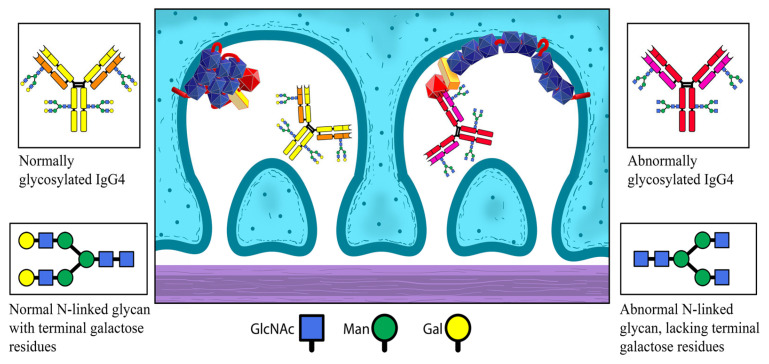
Lack of interaction between PLA2R and normally glycosylated IgG4; conformational changes in the structure of PLA2R after the activation of the lectin pathway and attachment of abnormally glycosylated IgG4.

**Figure 4 ijms-26-02206-f004:**
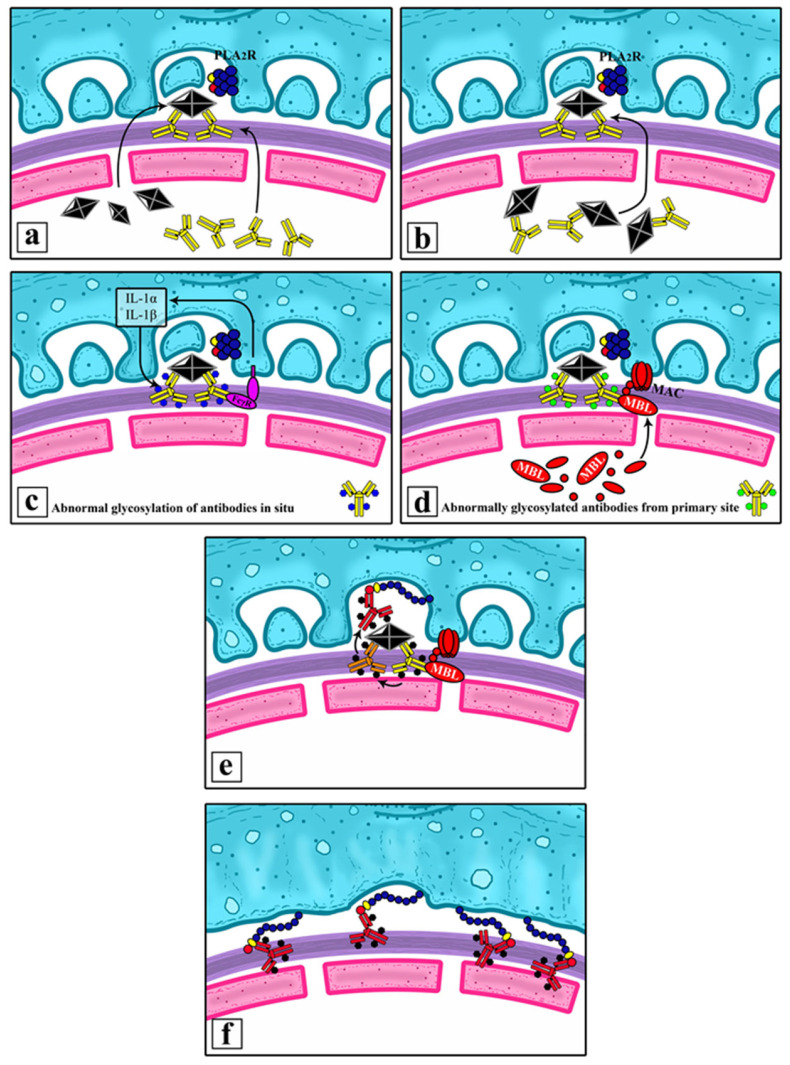
Hypothetical pathogenetic model. (**a**) Deposition of antigens and antibodies in the subepithelial space and formation of immune complexes in situ. (**b**) Deposition of preformed immune complexes in the subepithelial space. (**c**) Abnormal glycosylation of IgG4 under the influence of podocytes and activation of the lectin pathway. (**d**) Abnormal glycosylation of IgG4 in the primary site and activation of the lectin pathway in the kidney. (**e**) Sensitization of abnormally glycosylated antibodies from the immune complex against PLA2R that suffered conformational changes and is covering the immune complexes. (**f**) Sensitization of preformed antibodies against PLA2R from another site to PLA2R on the surface of podocytes.

**Figure 5 ijms-26-02206-f005:**
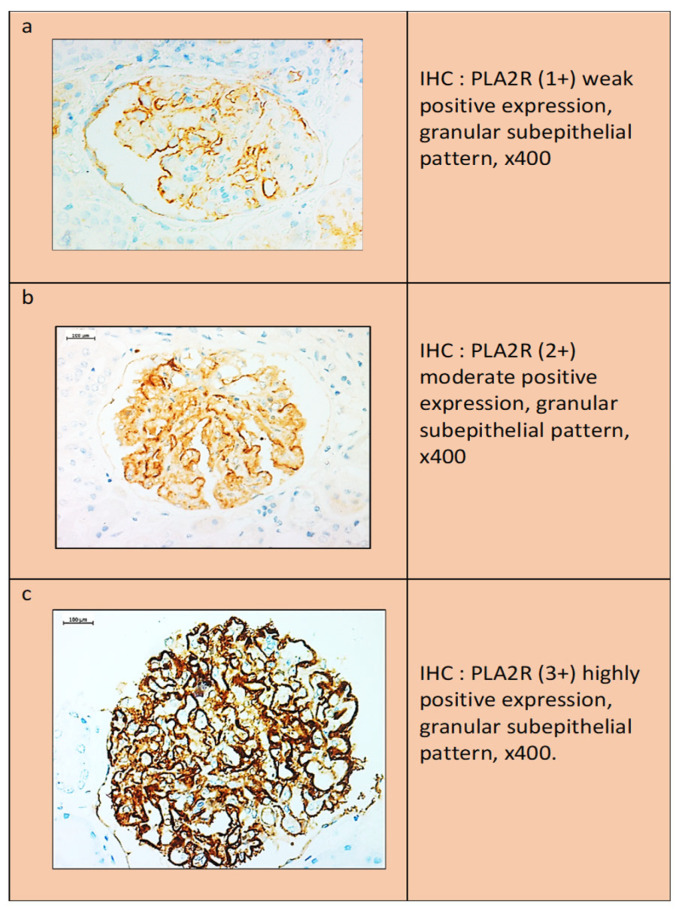
Microscopic images with their scoring of PLA2R expression. (**a**) Weakly positive PLA2R expression. (**b**) Moderately positive PLA2R expression. (**c**) Highly positive PLA2R expression.

**Table 1 ijms-26-02206-t001:** Patients with pMN and their comorbidities distributed by organs in which they are located.

Type of MN and General Data	Type of Comorbidity	Number of Patients	% in Relation to Total Number of Patients with pMN	Conclusion
pMN 69 patients 68% of the total number of patients with MN (102)	Thyroid gland	14	20%	In total, 69% of patients with pMN have/had a concomitant chronic inflammatory disease in a site, expressing PLAR2.
	Gall bladder	10	14%	
	Liver	9	13%	
	Lung diseases	15	22%	
	Non-relevant comorbidities	21	31%	In total, 31% of patients with pMN do not have relevant comorbidities.

**Table 2 ijms-26-02206-t002:** Patients with iMN and their comorbidities distributed by organs in which they are located.

Type of MN and General Data	Type of Comorbidity	Number of Patients	% in Relation to Total Number of Patients with iMN	Conclusion
iMN 23 patients 23% of the total number of patients with MN (102)	Thyroid gland	4	17%	In total, 91% of patients with iMN have/had a concomitant chronic inflammatory disease in a site, expressing PLAR2.
	Gall bladder	7	31%	
	Liver	1	4%	
	Lung diseases	9	40%	
	Non-relevant comorbidities	2	8%	In total, 8% of patients with iMN do not have relevant comorbidities.

**Table 3 ijms-26-02206-t003:** Patients with sMN and their comorbidities distributed by organs in which they are located.

Type of MN and General Data	Type of Comorbidity	Number of Patients	% in Relation to Total Number of Patients with sMN	Conclusion
sMN 10 patients 9% of the total number of patients with MN (102)	Liver	1	10%	In total, 20% of patients with sMN have/had a concomitant chronic inflammatory disease in a site, expressing PLAR2.
	Lung diseases	1	10%	
	Non-relevant comorbidities	8	80%	In total, 80% of patients with sMN do not have relevant comorbidities.

## Data Availability

The data presented in this study are available from the corresponding author upon request. The data are not publicly available due to national legal restrictions.

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
