# Peer review of "Hypothetical Pathogenetic Model of Membranous Nephropathy"

_ijms, 2025, doi:10.3390/ijms26052206_

Round 1

Reviewer 1 Report

Comments and Suggestions for Authors

Too many figures and too many tables. Consequently the manuscript is difficult to follow.

Authors should put together all the figures in a single table, for example. And select 2-3 figures, and put the rest of the figures in the supplementary material.

Introduction contains more than 2000 words. This is too long, and should be reduced to about 800 words.

Discussion. The following paragraph is not useful, and should be reduced in a single sentence.

“PLA2R1 is a receptor for secretory phospholipase A2 (sPLA2). The PLA2R gene is in Chromosome 2, Cytoband q24.2, Chromosome location (bp) 159932006 – 160062615. The encoded protein likely exists in both a transmembrane form and a soluble form. The transmembrane receptor may play a role in the clearance of phospholipase A2, thereby inhibiting its action. Polymorphisms at this locus have been associated with susceptibility to idiopathic membranous nephropathy. Alternatively spliced transcript variants encoding different isoforms have been identified [30,31].”

Several sentences in the discussion should be supported by a reference.

Discussion should be more focused on your results.

Author Response

Comments 1: Too many figures and too many tables. Consequently, the manuscript is difficult to follow.

Authors should put together all the figures in a single table, for example. And select 2-3 figures, and put the rest of the figures in the supplementary material.

Response 1: Thank you for pointing this out. We agree with this comment. Therefore, we have made some changes, the Tables with Schematic presentation of the data and diagnoses of patients will be united in one single table and will be downloaded in the supplementary material.

Comments 2: Introduction contains more than 2000 words. This is too long, and should be reduced to about 800 words.

Response 2: We agree and accordingly, reduced the introduction to 1344 words. We are presenting a Hypothetical Pathogenetic Model and we had to include what is discovered until that moment and reduced it to maximum. And still, names, dates and antibodies, although the abbreviations are consuming a lot of space. We believe they deserve to be mentioned in an expression of respect and gratitude for their discoveries.

Comments 3: Discussion. The following paragraph is not useful, and should be reduced in a single sentence.

“PLA2R1 is a receptor for secretory phospholipase A2 (sPLA2). The PLA2R gene is in Chromosome 2, Cytoband q24.2, Chromosome location (bp) 159932006 – 160062615. The encoded protein likely exists in both a transmembrane form and a soluble form. The transmembrane receptor may play a role in the clearance of phospholipase A2, thereby inhibiting its action. Polymorphisms at this locus have been associated with susceptibility to idiopathic membranous nephropathy. Alternatively spliced transcript variants encoding different isoforms have been identified [30,31].”

Response 3: We agree and accordingly, reduced it.

Comments 4: Several sentences in the discussion should be supported by a reference.

Response 4: We agree and reference are added.

Comments 5: Discussion should be more focused on your results.

Response 4: I agree and have made changes in the discussion. The number of patients with immunohistochemistry from site different from the kidney is small because not all of the patients had thyroidectomy and cholecystectomy in our hospital. That is why we decided to make the article as hypothesis and unite what is known until now and what we discovered. Hopefully somebody else in future, will have bigger number of patients with material from thyroid gland, liver and gall bladder. Also we added data from 102 patients with MN and their comorbidities.

4. Response to Comments on the Quality of English Language

Point 1:

Response 1:    (in red)

5. Additional clarifications

Thank you for your review, it is important for if you find it interesting and relevant and thank you for your comments, they really made the article better!

Reviewer 2 Report

Comments and Suggestions for Authors

Dear authors,

The article is very interesting and relevant, but I would like to make a few comments.

  1. The introduction to the article is very long, but this article is not an overview.
  2. The examined patiens were evalauted for other diseases, causing MN, as HBV-hepatitis B virus, SLE-systemic lupus erythematosus, sarcoidosis, cancer, exposure drugs induced MN?
  3. The number of investigated subjects is very small and the results of their research are very different.
  4. The recognized classification of MN is as follows: primary (manifests itself in the absence of an established cause (primary MN), and secondary (as infections - e.g., HBV, HCV—hepatitis C virus; autoimmune diseases as systemic lupus erythematosus, Hashimoto’s thyroiditis, Sjogren’s syndrome; cancers; drug intoxication; air pollution, cigarette smoking.
  5. To say that there was a connection between MN and cholecystitis, thyroiditis, hepatitis is too bold, because the number of subjects is very small to establish a serious correlation.
  6. Not all circumstances that could lead to the occurrence of MN have been fully investigated in patients.
  7. The introduction is very broad and detailed, but the results and conclusion are insignificant

Author Response

Response 1: Thank you for pointing this out. We agree with this comment. Therefore, we have reduced the introduction to 1344 words. This happened because of the research topic, presenting a Hypothetical Pathogenetic Model and we had to include what is discovered until that moment/and it really is a lot/ and reduced it to maximum. And still, names, dates and antibodies, although the abbreviations consumed a lot of space, but we believe they deserve to be mentioned in an expression of respect and gratitude for their discoveries.

Comments 2: The examined patients were evaluated for other diseases, causing MN, as HBV-hepatitis B virus, SLE-systemic lupus erythematosus, sarcoidosis, cancer, exposure drugs induced MN?

Response 2: Yes, before kidney biopsy all the patients are examined for HBV, HCV, HIV, antibodies for SLE. Asked for exposure to drugs and toxins. If it is needed examined for drugs, but they did not have MN, and are not included in this article. If we have abnormal findings on physical examination or ultrasound, then X-ray and CT scans are made, if the anemia is severe or the patient has gastrointestinal complains then endoscopy is scheduled. We added that in materials and methods.

Comments 3: The recognized classification of MN is as follows: primary (manifests itself in the absence of an established cause (primary MN), and secondary (as infections - e.g., HBV, HCV—hepatitis C virus; autoimmune diseases as systemic lupus erythematosus, Hashimoto’s thyroiditis, Sjogren’s syndrome; cancers; drug intoxication; air pollution, cigarette smoking.

To say that there was a connection between MN and cholecystitis, thyroiditis, hepatitis is too bold, because the number of subjects is very small to establish a serious correlation. Not all circumstances that could lead to the occurrence of MN have been fully investigated in patients.

Response 3: Yes, I completely agree this is why I added the number of patients that undergo kidney biopsy, with result MN for the period December 2012 until march 2023, their medical data was evaluated and concomitant diseases noted in excel file, and then I noticed the frequency of thyroiditis and cholecystitis, this is why I decided to make IHC on thyroid gland and gall bladder, on the material available in our hospital. I will add that data to the article and hope it will improve it. It really is bold but my aim was to clarify how MN appear, not to make changes in our therapeutical approach or in classification. I just notice during the main visitation a few cases with full remission after cholecystectomy /both of them had diabetes/. This is a big period of time and the patients were also monitored for the occurrence of malignant disease over the years. If medically indicated, tumor markers were examined.

Comments 4: The introduction is very broad and detailed, but the results and conclusion are insignificant

Response 4: I agree with this comment. Therefore, new patient data is added in results, discussion and conclusions are updated.

4. Response to Comments on the Quality of English Language

Point 1:

Response 1:    (in red)

5. Additional clarifications

Thank you for your review, it is important for us that you find it interesting and relevant and thank you for your comments, they really made the article better!

Round 2

Reviewer 1 Report

Comments and Suggestions for Authors

I would remove from the abstract that membranous nephropathy has a "growing incidence".

Introduction contains more than 2000 words. This is too long, and should be reduced to about 800 words. The history of the discovery of the various antigens is largely beyond the scope of this topic. I had already pointed this out in the previous review, and it was not taken into consideration. I would like to point out that such a long introduction makes the article unreadable and if it is not changed it will be a reason to not accept the article.

Methods do not present information regarding: study design, data collection, inclusion and exclusion criteria, and approval by the ethics committee with approval number.

In the present form the manuscript is very difficult to follow, so please, at the next submission I would suggest to include the track-changes form and the clean form of the manuscript.

Author Response

Comments 1: I would remove from the abstract that membranous nephropathy has a "growing incidence".

Response 1: Removed.

Comments 2: Introduction contains more than 2000 words. This is too long, and should be reduced to about 800 words. The history of the discovery of the various antigens is largely beyond the scope of this topic. I had already pointed this out in the previous review, and it was not taken into consideration. I would like to point out that such a long introduction makes the article unreadable and if it is not changed it will be a reason to not accept the article.

Response 2: We reduced the introduction to about 900 words. We do not agree that history of the discovery of the various antigens is largely beyond the scope of this topic. Heyman nephritis leads to discovery of megalin and we described megalin spreading. In humans we also have PLA2R spreading, supported by studies in the introduction. We use the described antibodies for differentiation pMN from sMN. We explain in our hypothetical model why we have positive antibodies for pMN in patients with sMN, in introduction are mentioned the same findings from other studies. We believe that abnormal glycosylation is leading to activation of lectin pathway, and in two sentences marked when it appears.

Comments 3: Methods do not present information regarding: study design, data collection, inclusion and exclusion criteria, and approval by the ethics committee with approval number.

Response 3: Information regarding: study design, data collection, inclusion criteria are described in rows 123-147, the added patients are described in materials and methods, rows 540-560 and the second ethics committee approval was added in the last submission at the end where is the Institutional Review Board Statement, we will highlight them.

Comments 4: In the present form the manuscript is very difficult to follow, so please, at the next submission I would suggest to include the track-changes form and the clean form of the manuscript.

Response 4: We can submit only one form, and we will submit the one with track -changes, which can be enabled from your side, from the track-changes menu.

Comments 5:

Response 4:

4. Response to Comments on the Quality of English Language

Point 1:

Response 1:    (in red)

5. Additional clarifications

Thank you for your review and thank you for your comments, they really made the article better!

Round 3

Reviewer 1 Report

Comments and Suggestions for Authors Table 1 is very interesting. Congratulations.   Figure 3 and 4 should be included with the figure 5 in only one single figure.   Radice et al. described APLA2R positivity is detectable in 70% of patients with pMN and 28% with sMN [24]. In a recent systematic review on APLA2R positivity, detected by indirect immunofluorescence (exactly the method adopted by Radice et al), APL2R was detectable in 73% of patients with pMN and 14% with sMN  [Allinovi et al]. Authors could include the following reference and its reported values of  APLA2R positivity in pMN and sMN, removing Radice et al (whose data are included in this systematic review): Allinovi M, Lugli G, Rossi F, Palterer B, Almerigogna F, Caroti L, Antognoli G, Cirami C. Accuracy of serum PLA2R antibody detected by indirect immunofluorescence in diagnosing biopsy-proven primary membranous nephropathy: a single-center experience and a systematic review of the literature. J Nephrol. 2023 Mar;36(2):281-283. doi: 10.1007/s40620-022-01528-1.

Author Response

Response 1: Thank you!

Comments 2: Figure 3 and 4 should be included with the figure 5 in only one single figure.

Response 2: Figure 3,4 and 5 included in one single figure.

Comments 3: Radice et al. described APLA2R positivity is detectable in 70% of patients with pMN and 28% with sMN [24]. In a recent systematic review on APLA2R positivity, detected by indirect immunofluorescence (exactly the method adopted by Radice et al), APL2R was detectable in 73% of patients with pMN and 14% with sMN  [Allinovi et al]. Authors could include the following reference and its reported values of  APLA2R positivity in pMN and sMN, removing Radice et al (whose data are included in this systematic review): Allinovi M, Lugli G, Rossi F, Palterer B, Almerigogna F, Caroti L, Antognoli G, Cirami C. Accuracy of serum PLA2R antibody detected by indirect immunofluorescence in diagnosing biopsy-proven primary membranous nephropathy: a single-center experience and a systematic review of the literature. J Nephrol. 2023 Mar;36(2):281-283. doi: 10.1007/s40620-022-01528-1.

Response 3: The following reference and its reported values included.
